# RvS: What is Essential for Offline RL via Supervised Learning?

**Scott Emmons**[1], **Benjamin Eysenbach**[2], **Ilya Kostrikov**[1], **Sergey Levine**[1]
[1]UC Berkeley,      [2]Carnegie Mellon University
`emmons@berkeley.edu`

## Abstract

Recent work has shown that supervised learning alone, without temporal difference (TD) learning, can be remarkably effective for offline RL. When does this hold true, and which algorithmic components are necessary? Through extensive experiments, we boil supervised learning for offline RL down to its essential elements. In every environment suite we consider, simply maximizing likelihood with a two-layer feedforward MLP is competitive with state-of-the-art results of substantially more complex methods based on TD learning or sequence modeling with Transformers. Carefully choosing model capacity (e.g., via regularization or architecture) and choosing which information to condition on (e.g., goals or rewards) are critical for performance. These insights serve as a field guide for practitioners doing Reinforcement Learning via Supervised Learning (which we coin *RvS learning*). They also probe the limits of existing RvS methods, which are comparatively weak on random data, and suggest a number of open problems.

## 1 Introduction

Offline and off-policy reinforcement learning (RL) are typically addressed using value-based methods. While theoretically appealing because they include performance guarantees under certain assumptions [27], such methods can be difficult to apply in practice; they tend to require complex tricks to stabilize learning and delicate tuning of many hyperparameters. Recent work has explored an alternative approach: convert the RL problem into a *conditional*, *filtered*, or *weighted* imitation learning problem. This typically uses a simple insight: suboptimal experience for one task may be optimal for another task. By conditioning on some piece of information, such as a goal, reward function parameterization, or reward value, such experience can be used for simple behavior cloning [5, 6, 8, 10, 17, 23, 26, 29, 32, 34, 38]. We refer to this set of approaches as RL via supervised learning (RvS). These approaches commonly condition on goals [6, 18, 29] or reward values [5, 26, 38], but they can also involve reweighting or filtering [10, 26, 32, 34].

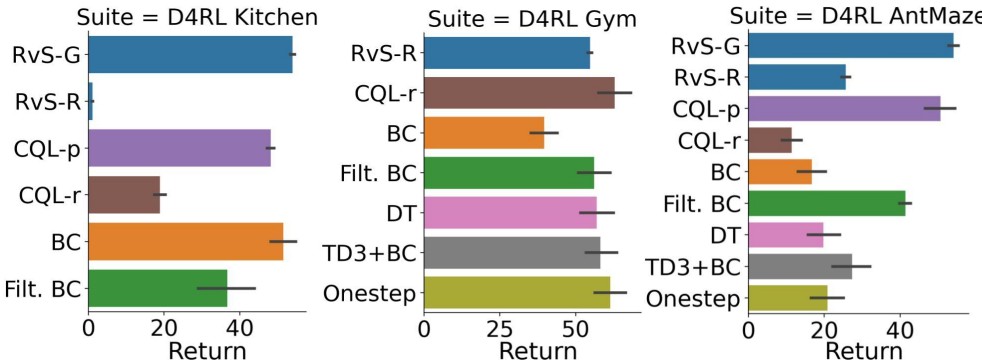

Figure 1: RvS learning conditioned on goals (RvS-G) and on rewards (RvS-R) compared with prior approaches and baselines. Each bar is the average over many tasks in each suite. Using just supervised learning with a feedforward MLP, RvS matches the performance of methods employing TD learning and Transformer sequence models. `https://github.com/scottemmons/rvs`

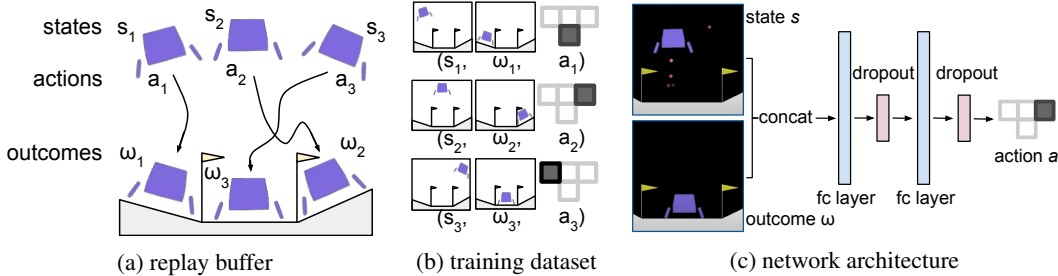

(a) replay buffer        (b) training dataset        (c) network architecture

Figure 2: (a) As input, RvS takes a precollected replay buffer of experience. An outcome $\omega$ can be an arbitrary function of the trajectory, such as future states or rewards. (b) RvS uses hindsight relabeling of the replay buffer to construct a training dataset. The observed actions act as demonstrations for the observed outcomes. (c) Our implementation of RvS uses an MLP with two fully connected (fc) layers to predict actions. At test time, we can condition on arbitrary outcomes.

RvS methods are appealing because of their algorithmic simplicity. However, prior work has put forward conflicting hypotheses about which factors are essential for their good performance, including online data [18], advantage weighting [26], or large Transformer sequence models [5]. The first question we study is: *what elements are essential for effective RvS learning?* Beyond this, it also remains unclear on which tasks and datasets such methods work well. For example, prior work has argued that temporal compositionality (dubbed "subtrajectory stitching") is an important component for solving offline RL when there are few near-optimal trajectories present in the data (e.g., the Franka Kitchen and AntMaze tasks in D4RL [12]). *A priori*, one might expect that dynamic programming via TD learning is needed for these tasks. So we also ask: *what are the limits of RvS learning, and does it scale to settings with few near-optimal trajectories?*

The main contribution of this paper is a study of RvS methods on a wide range of offline RL problems, as well as a set of analyses about which design factors matter most for such methods. First, we show that pure supervised learning (maximizing the likelihood of actions observed in the data) performs as well as conservative TD learning across a diverse set of environments. Second, simple feedforward models can match the performance of more complex sequence models from prior work across a wide range of tasks. Finally, choosing to condition on reward values versus goals can have a large effect on performance, with different choices working better in different domains. These simple results contradict the narrative put forward in many prior works that argue for more complex design decisions [5, 26, 27]. To the best of our knowledge, our results match or exceed those reported by any prior RvS method. We believe that these findings will be useful to the RL community because they help to understand the essential ingredients and limitations of RvS methods, providing a foundation for future work on simple and performant offline RL algorithms.

## 2    RELATED WORK

Recent offline RL methods use many techniques, including value functions [15, 27], dynamics estimation alongisde value functions [24, 37, 42], dynamics estimation alone [3, 7, 39], and uncertainty quantification [1, 41, 42]. In this paper, we focus on offline RL methods based on conditional behavior cloning that avoid value functions. The most common instantiation of these methods are goal-conditioned behavior cloning [8, 18, 29] and reward-conditioned behavior cloning [26, 35, 38]. Prior work has also looked at conditioning on different information, such as many previous timesteps [5, 23], tasks inferred by inverse RL [10], and other task information [6, 21]. While these methods can be directly applied to the offline RL setting, some prior works combine these methods with iterative data collection [18, 38], which is not permitted in the typical offline RL setting. Kumar et al. [26] study reward-conditioned policies and do present experiments in the offline RL setting. However, in contrast with our work, the results from Kumar et al. [26] suggest that good performance cannot be achieved through RvS learning but only through additional advantage-weighting [32, 34]. More recent work [5] presents a conditional imitation learning method for the offline RL setting that avoids advantage weighting by introducing a higher-capacity, autoregressive Transformer policy. In contrast to these prior works, we show that simple conditioning with standard feedforward networks can attain state-of-the-art results. While these results do not require advantage weighting or high-capacity sequence models, they do require careful tuning of the policy capacity.

## 3 REINFORCEMENT LEARNING VIA SUPERVISED LEARNING

In this section, we describe a formulation of Reinforcement Learning via Supervised Learning. We do not propose a new method but rather place many existing methods under a common framework. After this, we will investigate what design decisions are important to make such methods work well.

To formulate RvS methods in a general way, we assume an agent interacts with a Markov decision process with states $s_t$, actions $a_t$, initial state distribution $p_s(s_1)$, and dynamics $p(s_{t+1} \mid s_t, a_t)$. The agent chooses actions using a policy $\pi_\theta(a_t \mid s_t)$ parameterized by $\theta$. We assume episodes have fixed length $H$ and use $\tau = (s_1, a_1, r_1, s_2, a_2, r_2, \cdots)$ to denote a trajectory of experience. As some trajectories might be described using multiple outcomes, we use $f(\omega \mid \tau)$ to denote the distribution over outcomes that occur in a trajectory. We study two types of outcomes. The first type of outcome is a state ($\omega \in \mathcal{S}$) that the agent visits in the future: $f(\omega \mid \tau_{t:H}) = \mathrm{Unif}(s_{t+1}, s_{t+2}, \cdots, s_H)$. We refer to RvS methods that learn goal-conditioned policies as *RvS-G*. The second type of outcome is the average return ($\omega \in \mathbb{R}$) achieved over some future number of time steps: $f(\omega \mid \tau_{t:H}) = \mathbb{1}(\omega = \frac{1}{H-t+1} \sum_{t'=t}^{H} r(s_{t'}, a_{t'}))$. (Note that we found it important to use the max episode length as a constant $H$ in the denominator for all trajectories, treating early terminations as if they received 0 reward after they terminated.) We refer to RvS methods that learn reward-conditioned policies as *RvS-R*. It is also possible to condition on other information, such as the parameters of a reward function inferred via inverse RL [10] or the parameters of a task, such as turning left or right [6]. We focus on RvS methods applied to the offline RL setting. These methods take as input a dataset of experience, $\mathcal{D} = \{\tau\}$ and find the outcome-conditioned policy $\pi_\theta(a_t \mid s_t, \omega)$ that optimizes

$$\max_\theta \underbrace{\sum_{\tau \in \mathcal{D}}}_{\substack{\text{For all} \\ \text{trajectories:}}} \underbrace{\sum_{1 \le t \le |\tau|}}_{\substack{\text{For all timesteps} \\ \text{in that trajectory:}}} \underbrace{\mathbb{E}_{\omega \sim f(\omega|\tau_{t:H})}}_{\substack{\text{For all achieved} \\ \text{outcomes:}}} [\log \pi_\theta(a_t \mid s_t, \omega)].$$

While the basic formulation of this conditional policy is simple, instantiating RvS methods that attain excellent results has proven challenging [26, 38]. In the remainder of this paper, we present an empirical analysis of the design choices in RvS methods. We aim to understand which design choices are needed to make RvS methods perform well on diverse benchmark tasks, including how two different choices for $\omega$ (goals and rewards) compare in practice.

---

**Algorithm 1** RvS-Learning

1: **Input**: Dataset of trajectories, $\mathcal{D} = \{\tau\}$
2: Initialize policy $\pi_\theta(a \mid s, \omega)$.
3: **while** not converged **do**
4:     Randomly sample trajectories: $\tau \sim \mathcal{D}$.
5:     Sample time index for each trajatory, $t \sim [1, H]$, and sample a corresponding outcome: $\omega \sim f(\omega \mid \tau_{t:H})$.
6:     Compute loss: $\mathcal{L}(\theta) \leftarrow \sum_{(s_t, a_t, \omega)} \log \pi_\theta(a_t \mid s_t, \omega)$
7:     Update policy parameters: $\theta \leftarrow \theta + \eta \nabla_\theta \mathcal{L}(\theta)$
8: **end while**
9: **return** Conditional policy $\pi_\theta(a \mid s, \omega)$

---

## 4 TASKS AND DATASETS

To provide a comprehensive empirical study of RvS methods, we selected a broad range of tasks; the state-based tasks in prior work on various types of RvS or conditional imitation methods typically include only a subset of the domains we consider [5, 18, 26]. Our goal will be to include: *(1)* domains and datasets that are appropriate for different types of conditioning, including goals and rewards; *(2)* datasets that include different proportions of near-optimal data, ranging from near-expert datasets to ones with very few or no optimal trajectories at all; *(3)* datasets that run the gamut in terms of task dimensionality; *(4)* domains that have been studied by prior value-based offline RL methods.

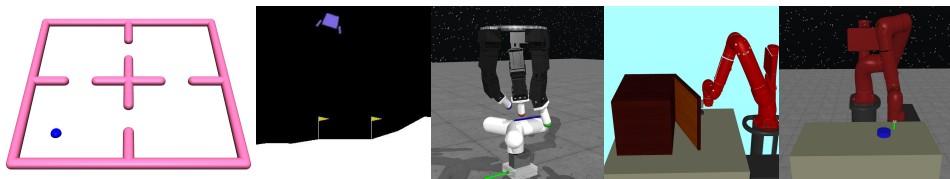

**GCSL** is a suite of goal-conditioned environments used by Ghosh et al. [18] to evaluate GCSL, a goal-conditioned RvS method with online data collection. We adapt these tasks for offline RL by using a *random* policy to collect training data, which results in suboptimal trajectories. The tasks include 2D navigation with obstacles (`FourRooms`, Eysenbach et al. [9]); two end-effector controlled Sawyer robotic arm tasks (`Door` and `Pusher`, Nair et al. [31]); the Lunar Lander video game, which requires controlling thrusters to land a simulated Lunar Excursion Module (`Lander`); and a manipulation task that requires rotating a valve with a robotic claw (`Claw`, Ahn et al. [2]).

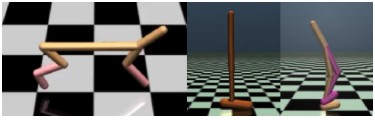

**Gym Locomotion** v2 tasks consist of the `HalfCheetah`, `Hopper`, and `Walker` datasets from the D4RL offline RL benchmark [12]. We use the `random`, `medium`, `medium-expert`, and `medium-replay` datasets in our evaluations, which consist of (a mixture of different) policies with varying levels of optimality. This requires learning from *mixed* and *suboptimal* data, and we will see that TD learning methods perform comparatively well on the `random` data.

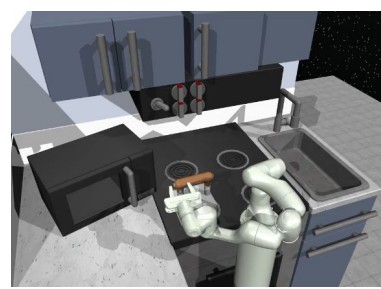

**Franka Kitchen** v0 is a 9-DoF robotic manipulation task paired with datasets of *human demonstrations*. This task originates from Gupta et al. [19] and was formalized as an offline RL task in D4RL [12]. Solving this task requires composing multi-step behaviors (e.g., open the microwave, then flip a switch) from component skills. This task includes three datasets: `complete`, where all trajectories solve all tasks in sequence; `partial`, where only a subset of trajectories perform the desired tasks in sequence; and `mixed`, which contains various subtasks but *never* all in sequence, requiring generalization and temporal composition.

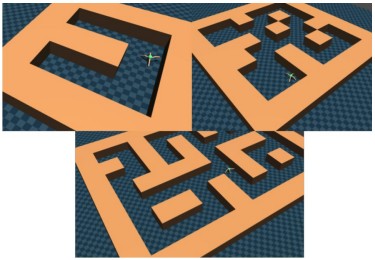

**AntMaze** v2 involves controlling an 8-DoF quadruped to navigate to a particular goal state. This benchmark task, from D4RL [12], uses a non-Markovian demonstrator policy and was intended to test an agent's ability to learn *temporal compositionality* by combining subtrajectories of different demonstrations. There are three mazes of increasing size: `umaze`, `medium`, and `large`, and there are two datasets types: `diverse` and `play`. The D4RL paper [12] claims that "diverse" data navigates from random start locations to random goal locations while "play" navigates between hand-picked start-goal pairs. Prior work has proposed that, for tasks requiring temporal compositionality, dynamic programming with value-based methods should be particularly important [12]. However, we find in AntMaze that RvS outperforms all the dynamic programming methods we consider.

When applying RvS learning methods, one must choose a type of outcome, such as rewards or goals. In Kitchen and AntMaze, the reward function is implemented in terms of goals. The kitchen reward is for completing multiple subtasks, so we condition RvS-G on a state in which all the subtasks have been achieved. For AntMaze, we condition RvS-G on the goal location in the maze. This choice makes the additional assumption that we know how the reward function is defined, rather than just receiving samples from the reward function. Because GCSL is inherently a multitask, goal-conditioned setup, we omit reward conditioning and only report RvS-G. Conversely, because there is no clear way to define performance in Gym w.r.t. a goal, we omit goal conditioning and only report RvS-R. For all tasks, we report scores in the range $[0, 100]$. For GCSL, the score corresponds to the percentage of episodes in which the agent achieves the goal. In the other environments, we follow Fu et al. [12] and normalize the score according to $100 \cdot \frac{\text{return} - \text{random}}{\text{expert} - \text{random}}$. (Images from [12, 18].)

## 5 ARCHITECTURE, CAPACITY, AND REGULARIZATION

We instantiate the RvS framework in Section 3 using a simple feedforward, fully-connected neural network. We condition on either the reward-to-go [26, 38] or a goal state [18, 19, 29] by merely concatenating with the input state (see Figure 2). Our aim is to identify the *essential* components of such methods; we eschew more complex architectures such as Transformer sequence models [5]. On

the tasks we consider, we show that our simple implementation achieves performance competitive with prior work that uses these more complex components. However, seemingly minor choices in the architecture do matter. These architectural choices include tuning capacity and regularization, suggesting that overfitting and underfitting are major challenges for RvS. Additionally, the choice of what to condition on (goals or rewards) also has important and domain-specific repercussions.

**Capacity and regularization.** In Figure 3, we compare different architecture sizes and regularization settings on a single task from each task suite. The best-performing architectures are generally larger than the architectures used in standard online RL [28] and imitation learning [11, 22]. This conclusion is intuitive, since RvS policies must represent both the optimal policy *and policies for other conditioning values* (e.g., other goals or suboptimal rewards). This result may help explain the contradictory conclusions from prior work regarding the importance of more complex design decisions [5, 26]. The results in Figure 3 also highlight the importance of regularization: while dropout sometimes has no effect (`pusher`), it improves performance in some tasks (`kitchen-complete`) and worsens performance in other tasks (`hopper-medium-expert` and `antmaze-medium-play`).

What can we say about these results on network capacity and regularization? The `hopper-medium-expert` dataset is large, and it contains only two modes: one from a medium-quality policy, and one from an expert policy. So, it is not surprising that this data doesn't need regularization. In contrast, `kitchen-complete` is a small dataset of human demonstrations solving many different subtasks. Prior work has found that human demonstrations are harder to fit than artificial demonstrations [30], and this may explain why `kitchen-complete` generally benefits from dropout. In `antmaze-medium-play`, relatively few trajectories are successful, so we are surprised to see that the best-performing hyperparameter setting does not use regularization. In `pusher`, regularization neither helps nor hurts, underscoring that the (lack of) impact of regularization depends on the task at hand. Overall, we speculate that regularization balances a tension between two competing demands on the policy. First, even if the behavior policy is simple, the *conditional* action distribution may be complex. Second, the policy must be sufficiently well regularized to generalize well to new goals or conditioning variables.

**Output distributions.** The form of the policy's output distribution also influences the policy's capacity. While unimodal Gaussians are a common choice with continuous action spaces [14, 26, 27], a categorical distribution over a discretized action space allows the policy to represent more complex, multi-modal distributions [18, 29]. To study the importance of the policy output distribution, we use the low-dimensional GCSL environments, where we can discretize the full action space. Figure 4 (left) shows that across the GCSL suite, categorical distributions either match or outperform Gaussian distributions. Similar to the results from the previous section, these results highlight how increasing the policy's model capacity can improve performance. Again, our finding stands in contrast to standard RL methods, which work well with unimodal Gaussians [20, 36] We conjecture that upweighting better data, as done by previous methods [26], may make it easier for low-capacity policies to fit the data. While both upweighting data and increasing model capacity allow the policy to better fit the data, increasing model capacity may be a simpler approach.

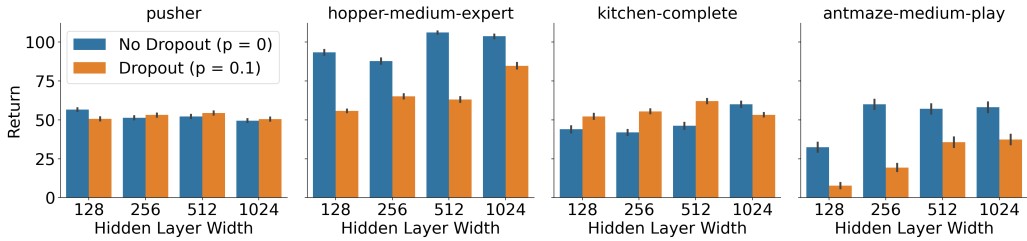

Figure 3: **Capacity and regularization**: We vary capacity (via network width) and regularization (via dropout) on one environment from each task suite. Larger networks perform better on `hopper-medium-expert` and `kitchen-complete`, suggesting the importance of high-capacity policy networks. However, dropout also usually boosts performance in `kitchen-complete`, suggesting that a combination of high-capacity policies with effective regularization is important for achieving good results.

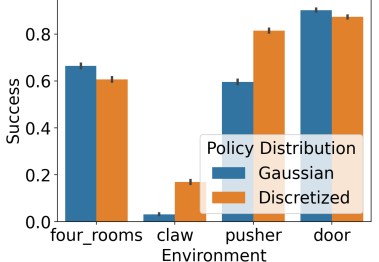 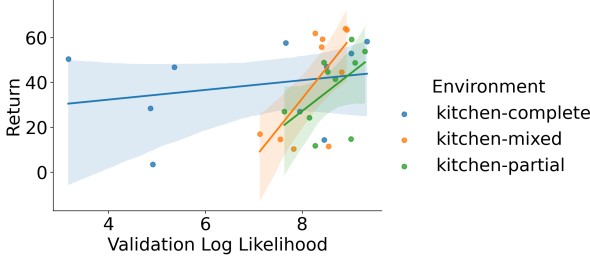

Figure 4: *(Left)* In the GCSL suite, a categorical distribution in discretized action space generally outperforms a unimodal Gaussian distribution in continuous action space. This fits the broad pattern we observe that higher-capacity RvS models perform better. *(Right)* In D4RL Kitchen, validation loss is only loosely correlated with final performance.

**Tuning with validation loss.** Hyperparameter optimization is especially important in the offline RL setting, as evaluating different hyperparameter configurations requires interacting with the environment [33]. Since RvS methods reduce the RL problem to a supervised learning problem, we hypothesize that the validation loss of this supervised learning problem might be an effective metric for tuning important hyperparameters, such as policy capacity and regularization. To test this, we train models on 80% of each Franka Kitchen dataset with two different hyperparameter settings: a regularized setting with dropout $p = 0.1$ and small batch size 256; and an unregularized setting without dropout and a large batch size of $16,384$. We show results in Figure 4 (right). For all three datasets, validation set error does correlate with performance, but the strength of this correlation varies significantly. In general, the validation loss does not provide a reliable approach for hyperparameter tuning. Fully automated tuning of hyperparameters remains an open question [13, 33].

**A recipe for practitioners.** We suggest the following process for online hyperparameter tuning: incrementally increase network width until performance saturates, and then try adding a bit of dropout regularization (e.g., dropout $p = 0.1$). If validation loss indicates that under/overfitting is an issue, one can also try increasing/decreasing the batch size, respectively.

Table 2 in the Appendix summarizes the hyperparameters that we found to work best for each task.

## 6 COMPARING RvS WITH PRIOR OFFLINE RL METHODS

Having identified key design decisions for implementing RvS learning, we now compare RvS to prior offline RL methods. We evaluate both goal-conditioned behavioral cloning (RvS-G) and reward-conditioned behavioral cloning (RvS-R) using the domains in Section 4. We then discuss what these results imply about the performance, design parameters, and limitations of RvS methods.

**Baselines and prior methods.** On the D4RL benchmarks, we compare RvS learning to *(i)* value-based methods and *(ii)* prior supervised learning methods and behavioral cloning baselines. For *(i)*, we include CQL [27] as well as the more recent TD3+BC [14] and Onestep RL [4] methods. TD3+BC and Onestep RL both involve elements of behavior cloning. We use CQL-p to denote the CQL numbers published in [12] and CQL-r to denote our best attempt to replicate these results using open-source code and hyperparameters from the CQL authors. For *(ii)*, we include behavioral cloning (BC), which does not perform any conditioning; Filtered ("Filt.") BC, a baseline appearing in [5] which performs BC after filtering for the trajectories with highest cumulative reward; and Decision Transformer (DT) [5], which conditions on rewards and uses a large Transformer sequence model. Both BC and Filtered BC are our own optimized implementations [25], tuned thoroughly in a similar manner as RvS-R and RvS-G. In Kitchen and Gym locomotion, Filtered BC clones the top 10% of trajectories based on cumulative reward. In AntMaze, Filtered BC clones all successful trajectories (those with a reward of 1) and ignores the failing trajectories (those with a reward of 0). On the GCSL benchmarks, we compare to GCSL using numbers reported by the authors [18]. This gives the GCSL baseline the advantage of online data, whereas RvS uses only offline data. We do not run RvS-G on the Gym locomotion tasks, which are typically expressed as reward-maximization tasks, not goal-reaching tasks. For additional details about the baselines, see Appendix A.

Table 1: **Overall performance.** RvS conditioned on goal states (RvS-G) is state-of-the-art in the AntMaze, Kitchen, and GCSL suites. RvS conditioned on rewards (RvS-R) performs worse at these tasks, highlighting the importance of the conditioning variable. In Gym, RvS-R matches the performance of Decision Transformer while only using an MLP. *For the sake of completeness, we include CQL-p, TD3+BC, and Onestep numbers that use AntMaze version v0. See Appendix A for more discussion of this and other experiment details.

| Suite | Environment | BC | Filt. BC | TD3+BC | Onestep | CQL-r | CQL-p | DT | RvS-R | RvS-G | GCSL |
|---|---|---|---|---|---|---|---|---|---|---|---|
| D4RL AntMaze | umaze-v2 | 54.6 | 60.0 | **78.6** | 64.3 | 44.8 | **74.0** | 65.6 | 64.4 | 65.4 | |
| | umaze-diverse-v2 | 45.6 | 46.5 | 71.4 | 60.7 | 23.4 | **84.0** | 51.2 | 70.1 | 60.9 | |
| | medium-play-v2 | 0.0 | 42.1 | 10.6 | 0.3 | 0.0 | **61.2** | 1.0 | 4.5 | **58.1** | |
| | medium-diverse-v2 | 0.0 | 37.2 | 3.0 | 0.0 | 0.0 | 53.7 | 0.6 | 7.7 | **67.3** | |
| | large-play-v2 | 0.0 | **28.0** | 0.2 | 0.0 | 0.0 | 15.8 | 0.0 | 3.5 | **32.4** | |
| | large-diverse-v2 | 0.0 | **34.3** | 0.0 | 0.0 | 0.0 | 14.9 | 0.2 | 3.7 | **36.9** | |
| | antmze-v2 average | 16.7 | 41.4 | 27.3* | 20.9* | 11.4 | **50.6*** | 19.8 | 25.6 | **53.5** | |
| D4RL Gym | halfcheetah-random-v2 | 2.3 | 2.0 | 11.0 | 6.9 | **18.6** | | 2.2 | 3.9 | | |
| | hopper-random-v2 | **4.8** | 4.1 | **8.5** | 7.8 | 9.3 | | 7.5 | **7.7** | | |
| | walker2d-random-v2 | 1.7 | 1.7 | 1.6 | 6.1 | 2.5 | | 2.0 | -0.2 | | |
| | random-v2 average | 2.9 | 2.6 | **7.0** | **6.9** | 10.1 | | 3.9 | 3.8 | | |
| | halfcheetah-medium-replay-v2 | 36.6 | 40.6 | **44.6** | 42.4 | 47.3 | | 36.6 | 38.0 | | |
| | hopper-medium-replay-v2 | 18.1 | 75.9 | 60.9 | 71.0 | **97.8** | | 82.7 | 73.5 | | |
| | walker2d-medium-replay-v2 | 26.0 | 62.5 | **81.8** | 71.6 | 86.1 | | 66.6 | 60.6 | | |
| | medium-replay-v2 average | 26.9 | 59.7 | 62.4 | 61.7 | **77.1** | | 62.0 | 57.4 | | |
| | halfcheetah-medium-v2 | 42.6 | 42.5 | 48.3 | **55.6** | 49.1 | | 42.6 | 41.6 | | |
| | hopper-medium-v2 | 52.9 | 56.9 | 59.3 | **83.3** | 64.6 | | 67.6 | 60.2 | | |
| | walker2d-medium-v2 | 75.3 | 75.0 | **83.7** | 85.6 | 82.9 | | 74.0 | 71.7 | | |
| | medium-v2 average | 56.9 | 58.1 | 63.8 | **74.8** | 65.5 | | 61.4 | 57.8 | | |
| | halfcheetah-medium-expert-v2 | 55.2 | **92.9** | **90.7** | 93.5 | 85.8 | | 86.8 | **92.2** | | |
| | hopper-medium-expert-v2 | 52.5 | **110.9** | 98.0 | 102.1 | 102.0 | | **107.6** | 101.7 | | |
| | walker2d-medium-expert-v2 | **107.5** | 109.0 | 110.1 | 110.9 | 109.5 | | 108.1 | 106.0 | | |
| | medium-expert-v2 average | 71.7 | **104.3** | 99.6 | 102.2 | 99.1 | | 100.8 | 100.0 | | |
| | gym-v2 average | 39.6 | 56.2 | **58.2** | 61.4 | 63.0 | | 57.0 | 54.7 | | |
| D4RL Kitchen | kitchen-complete-v0 | **65.0** | 4.0 | | | 11.8 | 43.8 | | 1.5 | 50.2 | |
| | kitchen-mixed-v0 | 51.5 | 40.0 | | | 24.2 | 51.0 | | 1.1 | **60.3** | |
| | kitchen-partial-v0 | 38.0 | **66.0** | | | 20.8 | 49.8 | | 0.5 | 51.4 | |
| | kitchen-v0 average | **51.5** | 36.7 | | | 18.9 | 48.2 | | 1.0 | **54.0** | |
| GCSL | claw | | | | | | | | | 14.8 | **28.0** |
| | door | | | | | | | | | 95.8 | 28.0 |
| | four rooms | | | | | | | | | 63.0 | **81.0** |
| | lunar | | | | | | | | | 54.7 | **70.0** |
| | pusher | | | | | | | | | 81.8 | **83.0** |
| | gcsl average | | | | | | | | | **62.0** | **58.0** |

**Overall performance and comparison to prior methods.** Table 1 shows the results of this comparison, leading to several conclusions about how our RvS implementations compare to prior value-based and supervised methods. In all suites, either RvS-G or RvS-R attains results that are comparable to the best prior method. The finding that conditional imitation with standard fully connected networks attains competitive results stands in contrast to prior work, which emphasizes advantage weighting and Transformer sequence models [5, 26]. As we discussed in Section 5, good performance requires a careful combination of high capacity and regularization, and it may simply be that this seemingly contradictory combination of components was not adequately explored in prior work.

**Subtrajectory stitching.** The `mixed` dataset in Kitchen and all of the AntMaze tasks consist of suboptimal trajectories. Attaining expert performance requires recombining parts of these trajectories. Such stitching is usually viewed as a key benefit of dynamic programming methods, so it is surprising that RvS-G performs as well as prior methods based on dynamic programming (TD3+BC and CQL). In these same tasks, RvS-G outperforms RvS-R. We speculate that the spatial information encoded by goal information helps RvS-G generalize. BC works well in the Kitchen environment but poorly in the AntMaze tasks, perhaps becaue the Kitchen task contains experience that is easier to imitate. As a direction for future work, we propose studying if conditioning on goals can help provide compositionality in *space* just as the Bellman backup provides compositionality in *time*.

**Reaching arbitrary goals.** An important capability of RvS-G, compared with more standard offline RL methods, is that it learns a policy that can reach many goals. To test how well RvS-G can learn to reach arbitrary goals from random offline data, we compare to GCSL, an RvS learning method that performs *iterative* online data collection [18]. We use the same tasks used by GCSL. Each of the tasks in the GCSL suite requires that the agent reach a goal drawn uniformly at random from the set of possible goals. The results shown in Figure 5 indicate that RvS-G can successfully reach many different goals entirely from randomly collected offline data. Although it may still be that iterative online collection, as proposed in prior work [18, 38], may be helpful in some domains, in this case simple offline training appears to suffice.

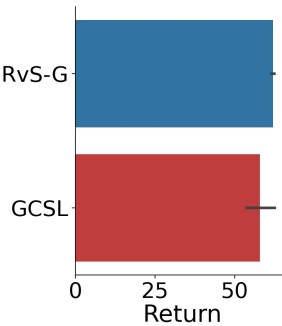

Figure 5: **GCSL results.**

**Random datasets.** While on average RvS-R is competitive with prior methods on the Gym Locomotion tasks, TD learning performs better on the random datasets; indeed, CQL performs especially well on `halfcheetah-random` (Table 1). This suggests that RvS methods may more generally perform worse on random datasets in comparison to TD learning methods.

**Analysis of reward conditioning.** Next, we analyze what RvS-R actually learns and how it uses the conditioning variable. First, we use the `walker2d-medium-expert` task to analyze the behavior of RvS-R for target rewards. The `walker2d-medium-expert` dataset contains two modes ("medium" and "expert", as shown in Figure 6). Conditioning the network on a range of reward targets (X-axis), Figure 6 shows that the policy's achieved return (Y-axis) corresponds only to the two modes in the dataset. The policy cannot interpolate between the two modes. In effect, it appears that RvS is mimicking a subset of the demonstrations, but doing so without explicit filtering (as is done in Filtered BC).

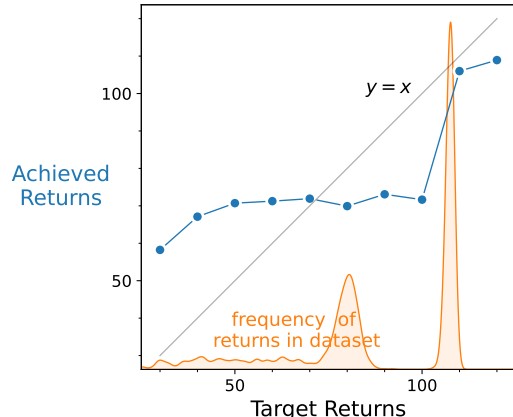

Figure 6: **A failure to interpolate.**

This analysis highlights the importance of the conditioning variable. It is not obvious *a priori* that a reward target of 100 will lead to a return of 70, whereas a reward target of 110 will lead to a return of 105. Following prior work [5], we tune the reward target for every task (see Table 2). Choosing this hyperparameter in a truly offline setting is an important problem for future work.

# 7 DISCUSSION AND FUTURE WORK

We presented an empirical study of offline RL via supervised learning (RvS) methods, which solve RL problems via conditional imitation learning. While prior work is divided on which elements of RvS are crucial for good performance, or even how well such methods actually work, our results provide three conclusions. *First*, if capacity and regularization are chosen correctly, RvS methods with simple fully connected architectures can match or outperform the best prior methods. *Second*, properly choosing the conditioning variable (e.g., goal or reward) is critical to the performance of RvS methods. *Third*, RvS methods remain competitive in tasks such as Franka Kitchen and AntMaze, where there is little optimal data. These conclusions suggest multiple directions for future work. Because policy capacity and regularization are critical to good performance, is there a method for automatically tuning these hyperparameters? Validation error is not a reliable metric. Also, the choice of conditioning variable is important: how might we automate this choice? Addressing these questions may increase the performance and applicability of RvS learning methods.

## ACKNOWLEDGMENTS

This work was supported in part by the DOE CSGF under grant number DE-SC0020347.

We thank Michael Janner for discussions about architectures, and we thank Aviral Kumar and Justin Fu for discussions about datasets. We thank Sam Toyer, Micah Carroll, and anonymous reviewers for giving feedback on initial drafts of this work.

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

Table 2: **Hyperparameters.** Architecture and design parameters that we found to work best in each domain. We define an epoch length to include all start-goal pairs in GCSL, i.e., to be $|\mathcal{D}|\binom{H}{2}$. In D4RL, we set all epoch lengths at 2000 $\binom{50}{2} = 2450000$.

| Hyperparameter | Value | Environment |
|---|---|---|
| Hidden layers | 2 | All |
| Layer width | 1024 | All |
| Nonlinearity | ReLU | All |
| Learning rate | 1e-3 | All |
| Epochs | 10 | GCSL |
| | 50 | Kitchen |
| | 2000 | Gym |
| Gradient steps | 20000 | AntMaze |
| Batch size | 256 | GCSL, Kitchen |
| | 16384 | Gym, AntMaze |
| Dropout | 0.1 | GCSL, Kitchen |
| | 0 | Gym, AntMaze |
| Goal state | Given | GCSL |
| | All subtasks completed | Kitchen |
| | (x, y) location | AntMaze |
| Reward target | 110 | Gym medium-expert, AntMaze, Kitchen |
| | 90 | Gym {hopper, walker2d}-medium-replay, walker2d-medium |
| | 60 | Gym hopper-medium |
| | 40 | Gym random, halfcheetah-{medium, medium-replay} |
| Policy output | Discrete categorical | GCSL |
| | Unimodal Gaussian | Kitchen, Gym, AntMaze |

## A  EXPERIMENT DETAILS

In this section, we provide more details about our experiments in each environment suite. For every task, we use 5 random training seeds and 200 evaluation rollouts for RvS.

**GCSL** Following the GCSL protocol [18], we collect offline data for RvS by performing random rollouts in each environment. As in GCSL, we set the environment time limit (i.e., max episode duration) to 50 actions, and the rollout policy uses a discretized action space. We collect 50,000 timesteps of experience in `FourRooms`, 100,000 timesteps of experience in `Door` and `Lander`, and 500,000 timesteps of experience in `Pusher` and `Claw`. We take our GCSL baseline numbers from those reported in the GCSL paper at the corresponding number of timesteps for each environment [18].

**Gym Locomotion** We use the v2 versions of the datasets for all algorithms. Because the v2 datasets fix issues with the v0 datasets, we omit the published CQL-p numbers, which use v0 and are no longer comparable. We instead report our replication of CQL, denoted CQL-r, using v2. We report the DT [27], TD3+BC [14], and Onestep [4] numbers from the corresponding papers. We use the reverse KL regularization variant of Onestep because this variant attains the best performance. Because DT did not run experiments with the random datasets, we run them ourselves using the default hyperparameters in the open-source codebase released by the DT authors. We omit numbers for RvS-G because there is no clear way to define the Gym tasks in terms of goals.

**Franka Kitchen** We use the v0 versions of the datasets and omit TD3+BC, Onestep, and DT numbers, which are not provided by the authors. We include the previously published CQL-p numbers from D4RL [12] as well as CQL-r, our own replication of CQL, and BC and Filtered BC, our own baselines.

**AntMaze** We use the v2 versions of the datasets for RvS, CQL-r, BC, Filtered BC, and DT. Because the DT authors do not test in AntMaze, we ran these experiments ourselves using the default hyperparameters in the open-source codebase released by the DT authors. We include previously published numbers for CQL-p [12] and TD3+BC [14], as well as Onestep numbers we received via email correspondence with the Onestep authors. These numbers for CQL-p, TD3+BC, and Onestep use the v0 versions of the datasets, which are the same as the v2 versions except that the v0 timeout

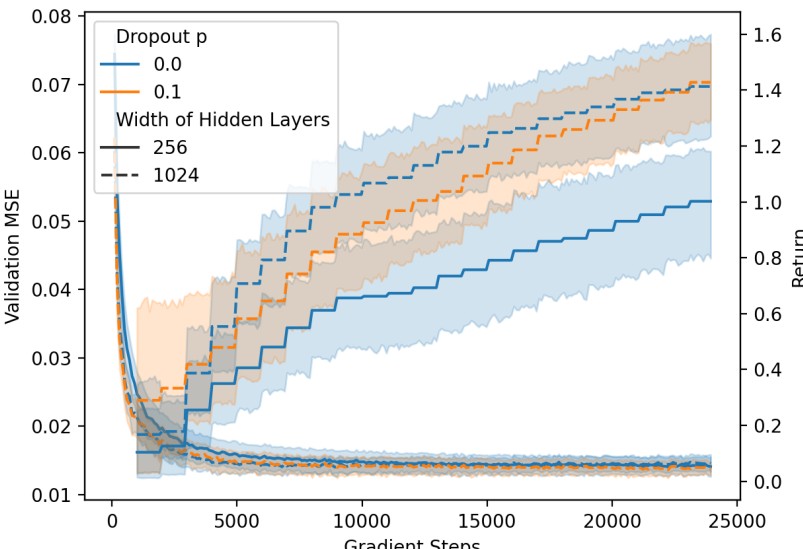

Figure 7: Validation mean-squared-error and evaluation return versus training gradient steps in Frank Kitchen. Each line includes 5 random seeds across each of the three datasets: `complete`, `partial`, and `mixed`. Surprisingly, we find that although each hyperparameter setting has nearly identical validation loss, the evaluation return varies by as much as 1.4x. This indicates that dropout and network capacity are having important effects on performance beyond validation loss.

flags indicating the ends of episodes contain errors. Because our initial experiments suggest that TD learning methods are fairly robust to errors in the timeout flags, we include these numbers for the sake of completeness.

**Codebases** Here are links to all of the codebases we use in our experiments:

1. RvS: `https://github.com/scottemmons/rvs`
2. CQL-r: `https://github.com/scottemmons/youngs-cql`
3. DT: `https://github.com/scottemmons/decision-transformer`
4. BC and Filtered BC: `https://github.com/ikostrikov/jaxrl`

## B  THE IMPACT OF MODEL CAPACITY AND REGULARIZATION

As we find in Section 5 that model capacity and regularization are important for performance in seemingly contradictory ways, we hypothesize that this can be explained by validation loss. So, for 5 random seeds in each of the 3 Franka Kitchen datasets (`complete`, `partial`, and `mixed`), we measure validation loss and return throughout training for three different settings of policy hyperparameters: network width 1024 with dropout $p = 0.1$, network width 1024 with dropout $p = 0$ (no dropout), and network width 256 with dropout $p = 0$ (no dropout). We expect that a larger network width leads to better validation loss, explaining its better return. We expect adding regularization also has this pattern of better validation loss and therefore better return.

In Figure 7, we plot the results: validation loss is nearly identical across all of the hyperparameter settings, but return can vary by as much as 1.4x between different hyperparameter settings. We also observe that evaluation return steadily increases even after validation loss has mostly leveled off. This is surprising and refutes our hypothesis; furthermore, it provides the insight that model capacity and dropout are having effects beyond just the validation loss. A similar phenomenon occurs in Neural Machine Translation, where cross entropy loss can decline while the quality of the translation

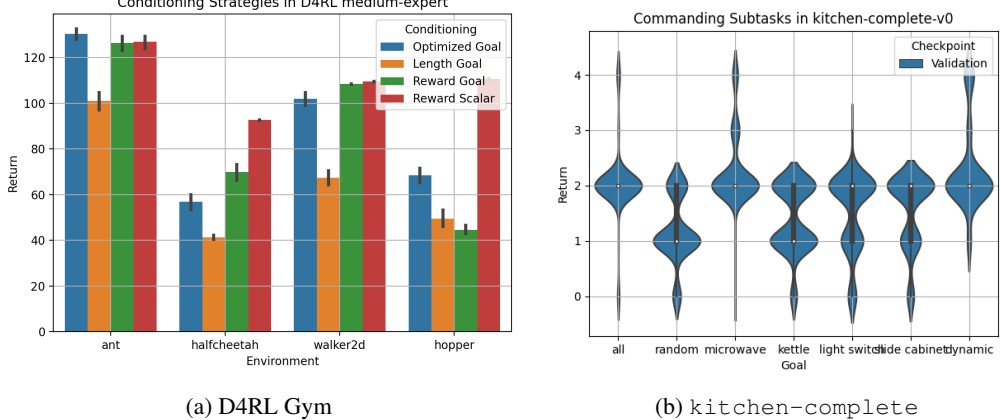

(a) D4RL Gym                    (b) `kitchen-complete`

Figure 8: (a) Different strategies for choosing goals in D4RL Gym. (b) Commanding the policy to complete different subtasks in `kitchen-complete`.

(measured by BLEU score) remains constant [16], and in contrastive representation learning, where lower capacity models have worse losses on the test set but still learn better representations for downstream tasks [40]. It is an important open problem for the field to understand why architecture decisions can impact performance beyond validation loss.

## C COMPARISON OF CONDITIONING STRATEGIES

To further study how different goal and reward conditioning strategies compare, in Figure 8a, we show the performance of four strategies: `Reward Scalar`, `Reward Goal`, `Length Goal`, and `Optimized Goal`. In Reward Scalar, we condition RvS-R on a return that is various fractions of expert performance. In Reward Goal, we condition RvS-G on states from randomly selected timesteps near the end of the highest-reward demonstration trajectories. In Length Goal, we condition RvS-G on states from randomly selected timesteps near the end of the longest demonstration trajectories. In Optimized Goal, we do a random search over 200 Length Goals and condition RvS-G on the highest performing one. (Note that Optimized Goal requires access to the environment, so it is not strictly offline RL; we are including it here as an oracle experiment.) We see that across all the goal selection strategies we consider, the performance of RvS-G cannot match the performance of RvS-R, since the Gym tasks are not inherently goal-conditioned.

In Figure 8b, we compare the performance of RvS-G in `kitchen-complete` when commanded to reach different goal states. The `kitchen-complete` task requires the agent to open a microwave, move a kettle, flip a light switch, and slide open a cabinet. For "all," we command the policy to reach a state where all the subtasks are achieved. For "random," we randomly select one of the tasks and command the policy to reach a state where that task is complete. For "microwave," "kettle," "light switch," and "slide cabinet," we command the agent to reach a state where that one subtask is achieved. For "dynamic," rather than leaving the goal fixed throughout the trajectory, we initially command the agent to complete the first subtask. Then, once the first subtask is complete, we update the command to include the second subtask (and so on for all remaining subtasks). As expected, we see that the oracle dynamic strategy, as well as conditioning on all subtasks completed, perform the best. Surprisingly, commanding the microwave to be opened outperforms commanding the other individual subtasks. We speculate that the microwave subtask might appear most often in the demonstrations.

