# OpenReview forum: "RvS: What is Essential for Offline RL via Supervised Learning?"
_ICLR.cc/2022/Conference — ICLR 2022 Poster_

### Official Review · Reviewer_oQXj · 2021-10-25

**Correctness:** 4
**Technical Novelty And Significance:** 1
**Empirical Novelty And Significance:** 2
**Recommendation:** 8
**Confidence:** 4

**Main Review:**

Strengths\
The paper provides a broad overview and makes a categorization.
Contributions to the discussion are provided, that might influence future research.

Weaknesses\
The finding that regularization of the learned policy is particularly significant suffers somewhat from the lack of a recipe to perform this regularization offline.

Other comments\
I find the sentence "For example, any experience is optimal for learning to reach the final state in that experience." too ambiguous.

The Introduction talks about a "reward parameter", I think at this point it should be made clearer what is meant by this.

"cloning(Codevilla" -> "cloning (Codevilla"

In the Related Work section, prior work is classified by whether it uses a value function, uses a dynamics model, or uses uncertainty quantification. It remains unclear that (Yu et al., 2020), (Shen et al., 2021) and (Kidambi et al., 2020) use both the dynamics model and a value function and only (Argenson & Dulac-Arnold, 2020) uses the dynamics model without a value function. At this point, it is helpful to also mention [1] and [2] as other examples of using dynamics models without using the value function.

[1] S. Depeweg et al., Learning and policy search in stochastic dynamical systems with Bayesian neural networks, ICLR 2017

[2] P. Swazinna et al., Overcoming model bias for robust offline deep reinforcement learning, EAAI 2021


"actions s_t" -> "actions a_t"

The axis labels in figures 1-5 are very small. It would be ideal if it had the same size as the caption font.

The text " RvS methods can often attain results that are comparable favorably to the best prior methods," should be corrected.

Please check the bibliography for accidental lower case letters, like „rl“, „q-learning“



**Summary Of The Paper:**

The paper investigates different variants of behavior cloning. The methods investigated aim at achieving better policies for use in offline RL through supervised learning than the average behavior contained in the data. The methods studied are categorized and examined against three to four benchmark problems.

**Summary Of The Review:**

The paper gives a good overview of different variants of behavior cloning. The contributions to the discussion may be helpful for future research. However, my feeling is that there is little reliable insight provided. Therefore, I think it is also possible that the benefit is small.

---

> ### Author Response · Authors · 2021-11-13
> **Author Response to oQXj**
>
> We thank the reviewer for their insightful feedback. Our understanding is that the reviewer is mainly concerned that we provide “little reliable insight.” However, we believe that our paper has considerable empirical significance. The particular design decisions we discuss _are_ simple and not particularly sophisticated, but that is the whole point: prior work has suggested that getting these types of conditional imitation (broadly, "RvS") methods to work requires a number of sophisticated choices and complex components. We show that this is not the case -- in fact, simply adjusting capacity carefully while using simple MLP models is sufficient. We believe this is very relevant to the community _in the context of prior work_, which has made different conclusions. Simplifications like this are often quite important to enable forward progress (see, e.g., [1-3]).
>
> Note also that we do an extensive empirical evaluation, considering all 26 environments of 4 diverse suites of tasks. When we added new experiments with the random and medium D4RL MuJoCo locomotion datasets as suggested by reviewer hoyq, we found that our general pattern of results continued to hold.
>
> **“lack of a recipe to perform this regularization offline”**
>
> An offline recipe for tuning hyperparameters for offline RL methods is a major open problem in the field [4, 5], so we unfortunately do not know how to provide this. However, we did add a subsection titled "A Recipe for Practitioners" to Section 5 that attempts to provide more general guidance about how practitioners can make use of our observations.
>
> **“I find the sentence… too ambiguous.”**
>
> We have removed this sentence from the paper.
>
> **“The Introduction talks about a "reward parameter", I think at this point it should be made clearer what is meant by this.”**
>
> We have changed this to “reward function parameterization.” Is it now clear?
>
> **“‘cloning(Codevilla’ -> ‘cloning (Codevilla’”**
>
> We have fixed this typo.
>
> **“Related Work section”**
>
> We have clarified which related work uses both a dynamics model and a value function and which uses the dynamics model without a value function. We have also added the references you suggest.
>
> **“actions s_t”**
>
> We have fixed this typo.
>
> **“small axis labels”**
>
> We will increase the font size of the axis labels.
>
> **“that are comparable favorably”**
>
> We have fixed this typo.
>
> **“check the bibliography for accidental lower case letters”**
>
> We have capitalized all acronyms in the bibliography that we can find. Please let us know if we’ve missed any.
>
> [1] Bello, Irwan, William Fedus, Xianzhi Du, Ekin D. Cubuk, Aravind Srinivas, Tsung-Yi Lin, Jonathon Shlens, and Barret Zoph. "Revisiting resnets: Improved training and scaling strategies." In Advances in Neural Information Processing Systems, 2021.
>
> [2] He, Kaiming, Ross Girshick, and Piotr Dollár. "Rethinking imagenet pre-training." In Proceedings of the IEEE/CVF International Conference on Computer Vision, pp. 4918-4927. 2019.
>
> [3] Xiao, Bo, Yuxuan Zhang, Yuan Chen, and Xianfei Yin. "FixMatch: Simplifying Semi-Supervised Learning with Consistency and Confidence." In Advances in Neural Information Processing Systems, 2021.
>
> [4] Fu, Justin, Mohammad Norouzi, Ofir Nachum, George Tucker, Alexander Novikov, Mengjiao Yang, Michael R. Zhang et al. "Benchmarks for Deep Off-Policy Evaluation." In International Conference on Learning Representations. 2020.
>
> [5] Paine, Tom Le, Cosmin Paduraru, Andrea Michi, Caglar Gulcehre, Konrad Zolna, Alexander Novikov, Ziyu Wang, and Nando de Freitas. "Hyperparameter selection for offline reinforcement learning." arXiv preprint arXiv:2007.09055 (2020).

---

> > ### Comment · Reviewer_oQXj · 2021-11-20
> > **Very good respone**
> >
> > I agree with the authors that demonstrating that simplifications are possible and that old familiar strategies are sufficient without the need for over-sophistication is very relevant. Additionally, the authors have addressed my concerns and suggestions very well. As it stands, I consider the paper to be an empirically strong, solid, and important work, that should be published to direct future research.

---

> ### Author Response · Authors · 2021-11-19
> **Checking In**
>
> Thank you again for your review. We hope you have had a chance to read our response. We would really appreciate a reply from you that lets us know if we have addressed your concerns and if there is anything else we can do to improve the paper.

---

### Official Review · Reviewer_dVtg · 2021-10-31

**Correctness:** 3
**Technical Novelty And Significance:** 3
**Empirical Novelty And Significance:** 3
**Recommendation:** 6
**Confidence:** 2

**Main Review:**

This paper provides empirical studies of offline RL via supervised learning.

## Strengthens

The biggest findings are two aspects:

* The authors find more complex design choices, such as the large sequence models and value-based weighting schemes used in some prior works, are generally not necessary, which is important in my opinion; especially, obtaining good performance using simple feedforward neural network is interesting;
* The empirical results show that carefully designed RvS methods can attain results that match or exceed the best prior methods across a range of different offline RL benchmarks, e.g. when goal or reward is carefully chosen.

## Weakness

Some of the expositions are not very clear:

1. The explanation of the objective (1) is not clear, for example, when $f$ uses RCBC, is $|f(\tau_{t+1}:H)|$ simply $1$? Also, why this objective should be used remain elusive.

2. Other than the goal and reward aspects, what are the other aspects that matter?










**Summary Of The Paper:**

This paper studies the importance of the design decisions for supervised learning type reinforcement learning algorithms. Through extensive experiments, find that more complex design choices, such as the large sequence models and value-based weighting schemes used in some prior works, are generally not necessary. Our results show that carefully designed RvS methods can attain results that match or exceed the best prior methods across a range of different offline RL benchmarks, including datasets with little or no optimal data.

**Summary Of The Review:**

In general, I like the new perspective this paper presented. In particular, the finding "find that more complex design choices are generally not necessary" provides a different way of thinking offline RL. Therefore, I prefer to accept at the moment.

---

> ### Author Response · Authors · 2021-11-13
> **Author Response to dVtg**
>
> We thank the reviewer for all of their comments. We understand the reviewer’s main concerns to be clarity about the objective function and understanding of what impacts performance beyond the goal / reward conditioning variable. To clarify the objective function, we have changed $f$ to be an expectation rather than a set (Section 3), and we have added annotation to each term in Equation 1. For analysis of other factors that impact performance, we have modified the opening paragraph of Section 5 to make it clear that the following paragraphs in that section describe other important aspects: capacity (network width), regularization (dropout), and output distributions (Gaussian vs discretized).
>
> **“when f uses RCBC, is |f(τt+1:H)| simply 1?”**
>
> Yes, this is correct. As this set notation for $f$ was unnecessarily confusing, we have revised the paper to describe the objective using an expectation over outcomes, rather than an average over a set.
>
> **“why this objective should be used remain elusive”**
>
> The objective is very simple -- it simply maximizes the likelihood of the dataset actions conditioned on the outcome that those actions achieve in the data. Equation 1 simply follows prior work [1]. Unfortunately, the notation we used in the original submission made this needlessly confusing, so we've revised it in the paper (see our updated Equation 1). We also added Figure 1 to illustrate the RvS method.
>
> **“Other than the goal and reward aspects, what are the other aspects that matter?”**
>
> We find that two aspects are model capacity and regularization. We study these aspects in Figure 2 by varying network size and dropout probability. We also find that a more expressive policy generally helps, as shown in Figure 3.
>
> Is there anything else you suggest we do to improve the quality of the paper?
>
> [1] Dibya Ghosh, Abhishek Gupta, Ashwin Reddy, Justin Fu, Coline Manon Devin, Benjamin Eysenbach, and Sergey Levine. Learning to reach goals via iterated supervised learning. In International Conference on Learning Representations, 2021.

---

> > ### Comment · Reviewer_dVtg · 2021-11-22
> > **Further response to authors**
> >
> > I thank the authors as they have addressed my concerns. However, I am not an expert in the empirical aspect of offline RL, so I will let other reviewers to make more decisive conclusions. I will keep the original score.

---

> ### Author Response · Authors · 2021-11-19
> **Checking In**
>
> Thank you again for your review. We hope you have had a chance to read our response. We would really appreciate a reply from you that lets us know if we have addressed your concerns and if there is anything else we can do to improve the paper.

---

### Official Review · Reviewer_hoyq · 2021-11-01

**Correctness:** 3
**Technical Novelty And Significance:** 2
**Empirical Novelty And Significance:** 2
**Recommendation:** 5
**Confidence:** 3

**Main Review:**

=== post author response: I have increased my score, but, I retain concerns about comparisons with more expressive classes of approaches such as the decision transformer.

1. While the paper presents interesting insights, I am not sure how much these findings can be generalized particularly in offline RL for locomotion tasks, where, typical benchmarks consider all 4 types of data collection strategies (random, medium, medium-replay, medium-expert) -- this paper considers only the latter two variants for their analysis. The other two variants are highly challenging in the offline RL context and require leveraging on elements of pessimism/conservatism to guide policy learning. The paper requires to present studies on these tasks to have a more holistic view of applying RvS methods to offline RL.

2. The comparison of this paper with the Decision Transformer is not well accounted for. The Decision Transformer work of Chen et al. (2021) can be viewed as a general framework used for synthesizing a variety of behaviors through conditioning on the rewards (or other events) during inference time. Such a perspective isn't quite considered by this work, and I am not sure how straightforward it is to programmatically generate a variety of behaviors without recourse to density modeling and sequence modeling. To present a fair comparison to works such as Decision Transformer, the paper needs to present results where a learnt policy can produce behaviors based on appropriate conditioning at inference time (and not just restricted to training time).

**Summary Of The Paper:**

This paper considers design choices involved in offline RL approaches that consider a reduction to weighted/conditional behavior cloning, a class of approaches referred to as Reinforcement Learning via Supervised Learning (RvS). The paper studies various issues including expressivity of policy architecture, regularization, choice of conditioning variables (e.g. based on goals or rewards).

At a high level, the takeaways are (a) RvS approaches are successful when using neural networks with appropriate capacity and regularization, (b) with appropriate conditioning variables (goal/reward based conditioning), (c) RvS can obtain policies that exhibit compositional behavior by conditioning on appropriate events.

**Summary Of The Review:**

The paper certainly takes a step forward with regards to understanding what components of RvS algorithmic frameworks matter to obtain satisfactory results - but, the paper falls short because (a) the paper doesn't present results with all types of datasets in the D4RL MuJoCo tasks (and this is standard in the offline RL literature), (b) the comparison to decision transformer isn't fully sketched out because the decision transformer framework is more general in that it offers the ability to generate a variety of behaviors at inference time by changing the conditioning variable during inference.

---

> ### Author Response · Authors · 2021-11-13
> **Author Response to hoyq**
>
> We thank the reviewer for their detailed review. We understand the reviewer's two main concerns to be (1) missing results on a subset of the D4RL datasets and (2) the relationship between RvS learning and Decision Transformer. We have run the additional experiments suggested by the reviewer and added results on the other D4RL locomotion datasets. We believe that the concern about Decision Transformers stems from a potential misunderstanding, which we explain below (and illustrate in Figure 6). **Do these improvements to the paper satisfactorily address the reviewer's concerns?**
>
> **“the paper doesn't present results with all types of datasets in the D4RL MuJoCo tasks”**
>
> Thank you for the excellent suggestion. We have now run RvS in the random and medium datasets of D4RL, and we also gather baseline numbers from the published work of the corresponding papers (with the exception of Decision Transformer, which doesn’t report numbers in D4RL random). See Table 2 for these updated results, which follow the same general pattern as before; RvS is generally competitive with all other algorithms in the D4RL random and medium datasets. However, we find that TD learning approaches generally have an edge of a few points in the random datasets, with CQL performing especially well in halfcheetah-random-v2. This generally aligns with our expectations, since random datasets provide the most potential for TD methods to improve over conditional behavioral cloning methods. In Section 6, we have added a paragraph called “Random datasets” that discusses this point.
>
> **“the Decision Transformer framework is more general [by] generat[ing] a variety of behaviors at inference time”**
>
> We believe that there may be a misunderstanding here: RvS-learning can also generate a variety of behaviors at inference time. For example, in the goal-conditioned tasks used in Figure 5 and Table 2, we evaluated the policy learned by RvS by commanding many different goals at inference time. Similarly, Figure 6 evaluates the policies learned by RvS by commanding many different rewards at inference time. To help clarify this in the paper, we have added the following text to the caption of Figure 1: “RvS can condition on arbitrary goals at both training and inference time.”
>
> Note also that we evaluate on the exact same D4RL tasks that the original Decision Transformer paper uses. However, if the reviewer believes that another comparison protocol would be more appropriate, please let us know!
>
> **Does this clarification address the reviewer's concerns about the relationship with Decision Transformer?** If not, or if we have misunderstood the reviewer's concerns, please let us know and we would be happy to further revise the paper or run additional experiments.

---

> ### Author Response · Authors · 2021-11-19
> **Checking In**
>
> Thank you again for your review. We hope you have had a chance to read our response. We would really appreciate a reply from you that lets us know if we have addressed your concerns and if there is anything else we can do to improve the paper.

---

> > ### Comment · Reviewer_hoyq · 2021-12-02
> > **Response to author comments**
> >
> > Thank you for your response. I have increased my score, but, i still retain some concerns about the reward conditioning angle.
> >
> > 1. With regards to the results in table 2 - thanks for the comparisons. I think this generally makes sense, I view the 10% BC to be highly competitive across tasks (and this somehow is not emphasized in much of literature), and, it does seem like that is the case with the table - which says 10% BC is indeed matching RvS methods in most of these tasks. Perhaps the authors can present a discussion surrounding this?
> >
> > 2. Wrt Reward conditioning - thanks for the clarification. It does seem like this plot highlights that RvS methods appear to mimic certain behaviors that exist in the offline dataset (as highlighted by the two modes result in figure 6) - something in principle that can be obtained by performing BC on the subset of data that obtains a specific target reward. Perhaps the authors can see why I am still unconvinced -- it does seem like in order to generate a highly diverse set of behaviors somewhat reliably, one does need to resort to more complex approaches such as the decision transformer (DT). From perspectives of presentation, I would recommend overlaying the results of the RvS methods on a plot like figure 4 in the decision transformer paper. I however see that in the walker case, DT does appear to generate a wider range of behaviors, though, correct me if I am wrong.

---

> > > ### Author Response · Authors · 2021-12-03
> > > **Responding to Follow-Up Comments**
> > >
> > > **“% BC somehow is not emphasized in much of literature”**
> > >
> > > Note that %BC is also competitive with Decision Transformer, and indeed the evaluation in the Decision Transformer paper (see Table 3) also finds that %BC is similar to DT in performance. So while the point about %BC being competitive is well-taken, and we will clarify and emphasize this, this was also found in the DT paper, published at NeurIPS.
> > >
> > > **“DT does appear to generate a wider range of behaviors”**
> > >
> > > Thank you for making this interesting observation. To further investigate this, we have run experiments where we condition RvS and Decision Transformer in the exact same way on a diversity of reward targets; here is [an anonymous link with the results](https://imgur.com/a/gslTGQ9). We find indeed that Decision Transformer is better at _interpolating_ intermediate reward values than RvS. This does suggest that Decision Transformer generates more diverse behavior, and we will add this figure and discussion to the paper.
> > >
> > > Nevertheless, we cannot think of a practical application where one would want to condition on less than optimal reward. We emphasize that Decision Transformer cannot _extrapolate_ to higher reward than RvS. **Both RvS and Decision Transformer attain the same maximum reward**, which is the primary evaluation criterion in both Decision Transformer and our work.

---

### Official Review · Reviewer_ehJE · 2021-11-03

**Correctness:** 3
**Technical Novelty And Significance:** 2
**Empirical Novelty And Significance:** 3
**Recommendation:** 6
**Confidence:** 3

**Main Review:**

It is good to see papers which deep dive into existing methods and study which aspects of the algorithm contribute to improved performance.

My major concern with the paper is the RvS algorithm is not described completely for me to understand the experiments. It appears that equation (1) in the paper is being used for gradient ascent for policy \pi parameterized with \theta. But the accumulation of rewards given by f() is in the denominator, so I do not understand how maximizing this objective leads to a policy that maximizes the rewards. The equation is unlike those given by papers cited (Kumar et al, Srivastava et al).

Given that this is supervised learning, the basic steps of training the model is also unclear. Essential hyper-parameter values such as learning rate is not provided. Given that there is no good way to determine overfitting as per validation set experiments in the paper, it is not clear how is training stopped in the first place. The number of epochs or steps of gradient update is not provided for any of the methods.

Given the above observation, it is difficult to believe the results reported in the paper. In addition, the most important insight from the paper is that model capacity and dropout play a key role in final performance. But, given these are contradictory, no further investigation is done to understand why this is the case. Instead, the paper leaves the subject with a conjecture.

The other aspects of the paper are not entirely surprising, and make sense.

**Summary Of The Paper:**

The paper studies the behavior cloning based strategies of offline RL algorithms in different type of environments and reports that performance primarily depends on model size and regularization. The results contradict some of the earlier claims, and the authors conjecture that model size and regularization characteristics can explain past results. The paper also discusses additional insights such as the importance of conditioning and simple validation based evaluation fail to generalize.

**Summary Of The Review:**

The algorithm under study is not described completely. Important hyper-parameters are missing. Because of this, the final results are circumspect.

---

> ### Author Response · Authors · 2021-11-13
> **Author Response to ehJE**
>
> We thank the reviewer for their detailed and insightful comments. It appears that the reviewer's two main concerns are the clarity of the method and experimental details. Below, we describe how we have addressed all the clarity issues mentioned by the reviewer. We have also newly uploaded supplemental material with code to reproduce the results, complementing the list of hyperparameters in Table 1. **Have these revisions to the paper addressed the reviewer's concerns about clarity and reproducibility?** We would be glad to incorporate additional suggestions on any part of the paper.
>
> **“the RvS algorithm is not described completely”**
>
> To clarify the description of RvS algorithms, we have added an algorithm block, Algorithm 1, that gives pseudocode for the method. We have also added an illustrative figure that gives a graphical depiction of RvS. Finally, we have added annotation to Equation 1 explaining what each term of the objective function is doing. Please let us know if there remains any confusion about the algorithm.
>
> **“The final results are circumspect…. Learning rate ... [and] number of epochs is not provided”**
>
> We agree that the original submission omitted some important hyperparameters. To correct this, we have added the missing hyperparameters (learning rate and epochs) to Table 1. We have also uploaded well-documented code as supplementary material to make reproducing the experiments easy. **Do these additional experimental details fully address the reviewer's concerns about the final results?** We would be glad to address other concerns as well, so it would be helpful if the reviewer would alert us of any remaining concerns.
>
> **“the accumulation of rewards given by f() is in the denominator”**
>
> To be clear, *the rewards were not in the denominator*. Rather, f() was a set of observed future outcomes, and |f()| was the cardinality of this set. We divided by |f()| because we were taking an expectation over future observed outcomes. In the case of RvS-R, there is just one future reward observed, so |f()| = 1.
>
> Your comment speaks to a broader issue: the set notation was confusing. To fix this, we have changed f() to be an expectation over outcomes (see Eq. 1). Furthermore, we have added annotation to Equation 1 to clarify what each term of the objective function is doing. Please let us know if any of the equations are still unclear.

---

> > ### Comment · Reviewer_ehJE · 2021-11-18
> > **Concerns addressed**
> >
> > The revised paper addresses majority of the concerns I have raised.
> >
> > Note that this particular comment was not responded to:
> > "the most important insight from the paper is that model capacity and dropout play a key role in final performance. But, given these are contradictory, no further investigation is done to understand why this is the case."
> >
> > In the revised paper, the following sentence is not clear to me:
> > "As some trajectories might be described using multiple outcomes"
> > My understanding is that each state visited in the trajectory can be treated as an outcome, and in the reward case, it is as if the trajectory were terminated early. It would be good to clarify if this is the case.

---

> > > ### Author Response · Authors · 2021-11-19
> > > **Responding to Follow-Up Comments**
> > >
> > > **“given [model capacity and dropout] are contradictory, no further investigation is done to understand why this is the case”**
> > >
> > > Thank you for the excellent suggestion. To investigate this, we have run an additional experiment with results given in a new figure, Figure 8. In particular, Figure 8 shows validation loss and evaluation return over the number of training gradient steps for different settings of model capacity and dropout. The individual validation loss curves are typical, showing a steep initial decline and then levelling off of validation loss. Similarly, the individual return curves are typical, showing steady increase in performance over the course of training. The interesting observation from the plot comes when we compare the curves to one another: we find that validation loss is nearly identical between different settings of model capacity and dropout while evaluation return can be 1.4x larger from one setting to another.
> > >
> > > This indicates that model capacity and dropout are impacting performance beyond validation loss. Similar behavior has been observed in other areas of deep learning. For example, in Neural Machine Translation, cross entropy loss can decline while the quality of the translation (measured by BLEU score) remains constant [1]. In contrastive representation learning, lower capacity models can have worse losses on the test set but still learn better representations for downstream tasks [2]. We believe that understanding the impact of network architecture beyond validation loss is an important avenue for future work.
> > >
> > > We have added a discussion of this to the paper under the heading “The Impact of Model Capacity and Regularization.”
> > >
> > > **“It would be good to clarify [how trajectories might be described using multiple outcomes]”**
> > >
> > > Thank you for bringing this to our attention. For RvS-G, each future state after time t in a trajectory is a possible outcome. For RvS-R, it is as if the trajectory were started late at time t, and the average cumulative reward is the single outcome. We have added this clarification to the caption of Figure 1.
> > >
> > > Please let us know if this addresses your concerns and if there is anything else we can do to improve the paper.
> > >
> > > [1] Ghorbani, Behrooz, Orhan Firat, Markus Freitag, Ankur Bapna, Maxim Krikun, Xavier Garcia, Ciprian Chelba, and Colin Cherry. "Scaling Laws for Neural Machine Translation." arXiv preprint arXiv:2109.07740 (2021).
> > >
> > > [2] Tschannen, Michael, Josip Djolonga, Paul K. Rubenstein, Sylvain Gelly, and Mario Lucic. "On Mutual Information Maximization for Representation Learning." In International Conference on Learning Representations. 2020.

---

### Author Response · Authors · 2021-11-13
**Author Response**

We thank the reviewers for all of their detailed comments and helpful suggestions. As we understand it, the reviewers’ main concerns are lack of clarity and details about the algorithm (ehJE and dVtg), lack of experiments in the random and medium datasets of D4RL MuJoCo locomotion (hoyq), and lack of a recipe for offline regularization (oQXj). While a recipe for offline hyperparameter tuning remains an open challenge for the entire field, we added a recipe for practitioners doing online tuning to Section 5, and we try to address all other concerns with the changes listed below:
1. We ran additional experiments that confirm that our results also hold with the random and medium D4RL Gym-MuJoCo locomotion datasets; see Table 2. (for hoyq)
2. We provide anonymized code and example scripts to run our experiments in newly uploaded supplemental material. (for ehJE)
3. We added learning rate and number of epochs to the hyperparameters of Table 1. (for ehJE)
4. We added an overview figure, Figure 1, that gives a graphical illustration to communicate the intuition behind our algorithm. (for ehJE and dVtg)
5. We added an algorithm block, Algorithm 1, to communicate our algorithm with step-by-step pseudocode. (for ehJE and dVtg)
6. We changed the mathematics of the outcome function f in Equation 1 to be in terms of an expectation over outcomes rather than a set of outcomes to prevent confusion from the set notation. (for ehJE and dVtg)
7. We have added annotation to Equation 1 to clarify what each term of the objective function is doing. (for ehJE and dVtg)
8. We updated the related work section, including two more relevant references. (for oQXj)
9. We corrected typos throughout the paper, including capitalization and grammar issues. (for oQXj)

Please see our individual responses to reviewers for more details.

**Have these changes addressed the reviewers’ concerns?** If any concerns remain, please let us know; we will be happy to address them.

---

### Decision · Program_Chairs · 2022-01-20

**Decision:**

Accept (Poster)

**Comment:**

The paper studies the behavior cloning based strategies of offline RL algorithms in different type of environments and reports that performance primarily depends on model size and regularization. The results contradict some of the earlier claims, and the authors conjecture that model size and regularization characteristics can explain past results.

During the review period, the reviewers agreed that the paper has certain merits, and on the other hand, they also raised some concerns, regarding some missing technical details, whether the empirical finding could be trusted, the generalization of the findings to more scenarios, and the comparison with some highly related papers. The authors did a good job in their rebuttal, which removed many of the above concerns (although not all) and convinced the reviewers to raise their scores. As a result, we believe it is fine to accept the paper (although somehow like a weak accept).